# Impact of COVID-19 on Breastfeeding among SARS-CoV-2 Infected Pregnant Women: A Single Centre Survey Study

**DOI:** 10.3390/ijerph20010228

**Published:** 2022-12-23

**Authors:** Angelo Sirico, Roberta Musto, Sonia Migliorini, Serena Brigidi, Federica Anzelmo Sciarra, Annunziata Carlea, Gabriele Saccone, Maurizio Guida, Laura Sarno

**Affiliations:** 1Department of Neuroscience, Reproductive Sciences and Dentistry, University of Naples Federico II, 80131 Naples, Italy; 2Department of Anthropology, Philosophy, and Social Work, University of Rovira i Virgili, 43003 Tarragona, Spain; 3Medical Anthropology Research Center, University of Rovira i Virgili, 43003 Tarragona, Spain

**Keywords:** COVID-19, breastfeeding, pregnancy, lactation, vertical transmission, SARS-CoV-2, breast milk, infection, psychological, survey

## Abstract

Background: Although current guidelines recommend that mothers with suspected or confirmed SARS-CoV-2 infection should be encouraged to initiate and continue breastfeeding, up-to-date literature shows conflicting data regarding breastfeeding experiences in infected women. This survey aimed to report on the psychological impact of SARS-CoV-2 infection on breastfeeding practice and medical counselling in a single tertiary center in Southern Italy. Methods: One-hundred breastfeeding women with SARS-CoV-2 infection at delivery were given an anonymous questionnaire regarding breastfeeding and women’s perception of the impact of COVID-19 on breastfeeding. Results: 75% of women reported they had difficulty breastfeeding; among them, 66 (66%) declared that separation from their babies after delivery affected their ability to breastfeed. Incidence of reported difficulties in breastfeeding was higher in women who underwent caesarean section compared to women with vaginal delivery (56/65, 86.2% vs. 19/35, 54.3%, χ^2^ = 12.322, *p* < 0.001) and in women with a hospital stay of more than 5 days (48/57, 84.2% vs. 23/37, 62.2%, χ^2^ = 5.902, *p* = 0.015). Furthermore, the incidence of difficulties in breastfeeding was higher in women who subsequently decided to use exclusively infant formula compared to women who mixed maternal milk with infant formula and women who breastfed exclusively with maternal milk (48/49, 98% vs. 20/25, 80% vs. 7/26, 26.9%, χ^2^ = 46.160, *p* < 0.001). Conclusions: Our survey highlights the importance of healthcare support and information on hygiene practices to decrease the perceived stress related to breastfeeding for infected mothers under restrictions, especially in women undergoing cesarean section and with a long hospital stay.

## 1. Introduction

Since March 2020, the SARS-CoV-2 pandemic has posed an outstanding emergency for national health systems worldwide; to date, nearly 600 million cases and 6 million deaths have been confirmed worldwide [1]. The severity of SARS-CoV-2 infection in the general population has been reported to be significantly influenced by different risk factors. Among these, age and comorbidities were found to be the strongest predictors of hospital admission, critical illness, and mortality [2]. In particular, pregnant women, especially in cases of high-risk pregnancies, have been found to carry a higher risk of composite adverse maternal outcomes, severe respiratory symptoms, and invasive ventilation compared to low-risk pregnancies [3]. According to current knowledge, pregnant women have an increased risk of developing severe pneumonia because they are more susceptible to respiratory pathogens due to changes in their immune systems [4]. A recent meta-analysis showed that black and Asian ethnicities, obesity, hypertension, and asthma are risk factors for developing COVID-19 related symptoms in pregnancy, although other authors report similar outcomes between COVID-19 positive and negative pregnant women, with complications (mostly bleeding or thrombosis) occurring only in a small percentage of COVID-19 cases [5,6].

The first recommendations suggested that suspected, probable, and confirmed cases of COVID-19 infection should be managed initially by designated tertiary hospitals with effective isolation facilities and protection equipment to assure fetal and maternal benefits along with minimal risks for healthcare providers [7,8]. As soon as the first infections of pregnant women with COVID-19 were reported, scientific organizations raised concerns about the safety of breastfeeding in these cases.

Breastmilk is rich in nutrients, including Igs, lactoferrin, human milk oligosaccharides, and anti-inflammatory factors [9] that are beneficial for newborns, while partial breastfeeding or non-breastfeeding increases the risk of neonatal diarrhea or respiratory infections and decreases infant survival rate [10]. Newborns or preterm infants lacking breastfeeding are prone to sudden infant death syndrome, necrotizing enterocolitis (NEC), sepsis [11], and even higher mortality than breastfed infants.

In March 2020, the WHO published a statement recommending that mothers with suspected or confirmed COVID-19 should be encouraged to initiate and continue breastfeeding, and that mothers should be counseled that the benefits of breastfeeding substantially outweigh the potential risks of transmission [12]. However, different countries provided national guidelines on breastfeeding in COVID-19 women, not always adhering to the WHO recommendations, and changing final recommendations over time. Furthermore, while some mothers reported positive breastfeeding experiences during the pandemic, many mothers indicated negative experiences related to decreased social and professional support. At present, available data on COVID-19-infected women are limited [13]. This study aimed to report the impact of the COVID-19 infection on women breastfeeding practice and medical counselling in a single tertiary center in Southern Italy.

## 2. Materials and Methods

A survey study was conducted at the COVID Unit of the Mother and Child Department at University Hospital Federico II in Naples, Italy, from November 2020 to May 2021. The aim of the survey was to gather data from SARS-CoV-2 infected mothers regarding their breastfeeding experience and women’s perception of the impact of COVID-19 on breastfeeding. All procedures performed in this study were in accordance with the ethical standards of the institutional and/or national research committee, and with the 1964 Helsinki declaration and its later amendments or comparable ethical standards. This study obtained the approval from the local IRB (Protocol #145/20, approved on 3 April 2020). Women were enrolled at admission to the Unit and informed consent was obtained before participating to the survey.

During the COVID-19 pandemic, infrastructural adaptations were made to provide an isolated route for infected patients, including a dedicated obstetric emergency room, a dedicated antenatal ward, a labor ward, an operating theatre, and a bio-containment area for suspected cases. Patients with a positive molecular nasal/oropharyngeal swab, which is a test that detects the genome (RNA) of the SARS-CoV-2 virus in the biological sample using the RT-PCR method, were admitted into the dedicated COVID-19 ward. We included in our analysis consecutive pregnant women who were diagnosed with SARS-CoV-2 infection and delivered at our Unit. Exclusion criteria were the presence of a positive rapid antigen-test without confirmed infection with a molecular test, persistent asymptomatic infection after 21 days from the first molecular test, infected patients who were discharged from our Unit before delivery, patients who were subsequently diagnosed negative after a molecular nasal/oropharyngeal swab before delivery, presence of fetal chromosomal or congenital anomalies, and admission of the newborn to the Neonatal Intensive Care Unit.

Included women were given an anonymous questionnaire regarding breastfeeding and women’s perception of the impact of COVID-19 on breastfeeding. An English version of the questionnaire is available as a Appendix A.

For each patient, we collected the following data: maternal age, parity, last menstrual period, education level, marital status, occupational status, fetal growth restriction (FGR), pregestational and gestational diabetes, maternal hypertension (including chronic hypertension, gestational hypertension, and preeclampsia), COVID-19 related symptoms, gestational age (GA) at delivery, and type of delivery (spontaneous delivery or cesarean section).

Descriptive analysis has been performed and data have been reported as a mean ± standard deviation for continuous variables, and as a number (percentage) for categorical variables. Differences between groups have been assessed by Student’s *t*-test or the Mann–Whitney test for continuous variables, and by Chi-squared test or Fisher’s exact test for categorical variables. Statistical analysis was carried out using the Statistical Package for Social Sciences (SPSS) Statistics v. 19 (IBM Inc., Armonk, NY, USA).

## 3. Results

### 3.1. Characteristics of Study Participants

One-hundred breastfeeding women with SARS-CoV-2 infection at delivery were included in the analysis. A detailed overview of the study participants’ characteristics is reported in Table 1.

The mean maternal age was 30.1 ± 6.1 years while the mean gestational age at delivery was 38.7 ± 1.9 weeks gestation. Only 7 women (7%) experienced a preterm birth. Most women were nulliparous (*n* = 53; 53%), housewives (*n* = 50; 50%), married (*n* = 58; 58%), and were asymptomatic for SARS-CoV-2 infection (*n* = 73; 73%). Only 27 women (27%) showed symptoms, including cough (*n* = 16; 16%), anosmia/dysgeusia/anorexia (*n* = 7; 7%), cold (*n* = 7; 7%), dyspnea (*n* = 6; 6%), fever (*n* = 5; 5%), and articular pain (*n* = 2, 2%). Only 2 cases (2%) of clinical pneumonia and 1 (1%) case of diarrhea were reported. Only 2 (2%) required oxygen therapy due to worsening of the respiratory symptoms. Sixty-five women delivered by cesarean section (65%) and the mean length of stay for hospitalization was 7.2 ± 3.5 days, while no neonatal infections were reported. All the included women were unvaccinated against SARS-CoV-2 since the vaccine was neither available nor recommended yet at time of the enrollment.

### 3.2. Effects of COVID-19 on Obstetricians or Other Medical Counselling during Pregnancy and Breastfeeding

In total, 92% of all pregnant women reported that their pregnancy was mainly followed-up by a private obstetrician, while a midwife or public obstetric clinic was involved in 8% of cases. Importantly, 17% reported having received less follow-up by obstetricians compared to before the pandemic and less medical counselling during the breastfeeding period, and they felt abandoned, neglected, or isolated. The coronavirus pandemic influenced their current pregnancy follow-up to some extent. According to the survey, 23% of included women reported that they did not receive information about how to reduce the risks of SARS-CoV-2 transmission during breastfeeding, while 12% of women were not informed about how to wash the breast before breastfeeding. Furthermore, the use of a breast pump was recommended in 60% of cases, while in 27% of cases breastfeeding was even advised against.

### 3.3. Breastfeeding Experience

According to our Institution’s policy, all mothers were separated from their babies after delivery due to COVID-19 positivity, however, no newborn tested positive at birth. Only 3 women (3%) had the possibility to have skin-to-skin contact after delivery. Breastfeeding characteristics are reported in Table 2.

Only 10 women (10%) reported to have breastfed during the hospital stay, using breast pumps, and in 2 of these cases maternal milk was mixed with infant formula, while 51 women (51%) declared they breastfed at home; 26 (26%) women breastfed exclusively with maternal milk, while 25 (25%) women reported they mixed maternal milk with infant formula. Only 19 women (19%) reported they breastfed directly to the baby without using breast pumps. Furthermore, the duration of breastfeeding was variable: 72.5% of them (29/40) breastfed for a period of 1 to 6 months, 17.5% (7/40) breastfed less than one month, and only 10% (4/40) over 7 months.

Regarding the influence of SARS-CoV-2 infection on breastfeeding, 75 women (75%) reported they had difficulty breastfeeding; among them, 66 (66%) declared that separation from their babies after delivery affected their ability to breastfeed, despite that 82.9% (39/47) of the multiparous mothers had previous breastfeeding experience.

Overall, 29/100 surveyed women reported that SARS-CoV-2 infection had a major influence on breastfeeding due to fear of transmission to their babies, while 14 women (14%) reported difficulties breastfeeding due to the severity of their infections. Of the included women, 25 (25%) reported that the distance from the baby for a long period, due to the infection consequences and the hospitalization, was the main cause for not having breastfed, while 18 women (18%) reported they experienced a reduced milk production due to the emotional impact of the pandemic and the fear of contagion. According to the survey, 3 women (3%) decided voluntarily not to breastfeed and reported they had not breastfed after their previous pregnancies.

Data analysis showed that there was no difference in reported difficulties in breastfeeding according to the incidence of preterm birth (5/7, 71.4% vs. 70/93, 75.3%, χ^2^ = 0.051, *p* = 0.821), the presence of COVID-19 related symptoms (22/27, 81.5% vs. 53/73, 72.6%, χ^2^ = 0.829, *p* = 0.363), or due to nulliparous status (38/53, 71.7% vs. 37/47, 78.7%, χ^2^ = 0.656, *p* = 0.418). Reported difficulties in breastfeeding were not different whether women correctly received information about how to reduce the risks of SARS-CoV-2 transmission during breastfeeding (57/77, 74% vs. 18/23, 78.3%, χ^2^ = 0.169, *p* = 0.681) or information on breast washing before breastfeeding (9/12, 75% vs. 66/88, 75%, χ^2^ = 0.000, *p* = 1.00). On the other hand, the incidence of reported difficulties in breastfeeding was higher in women who underwent caesarean section compared to women with vaginal delivery (56/65, 86.2% vs. 19/35, 54.3%, χ^2^ = 12.322, *p* < 0.001) and in women with hospital stay more than 5 days (48/57, 84.2% vs. 23/37, 62.2%, χ^2^ = 5.902, *p* = 0.015). This data is shown in Table 3.

When evaluating the relationship between reported difficulties in breastfeeding and the type of infant nutrition after hospital discharge, data analysis showed that the incidence of difficulties in breastfeeding was higher in women who subsequently decided to use exclusively infant formula compared to women who mixed maternal milk with infant formula and women who breastfed exclusively with maternal milk (48/49, 98% vs. 20/25, 80% vs. 7/26, 26.9%, χ^2^ = 46.160, *p* < 0.001), as shown in Figure 1.

## 4. Discussion

The current study aimed to assess the perceived impact of the SARS-CoV-2 pandemic on women and their breastfeeding practices, and on the level of medical advice received during pregnancy and breastfeeding. Our findings showed that 75% of women had difficulty breastfeeding during the pandemic, and 66% of them reported that COVID-19 had a major influence on breastfeeding, mostly due to fear of transmission to their children. Our survey highlighted that, in infected women, difficulties in breastfeeding were significantly associated with delivery by cesarean section and the duration of hospital stay, and that there was an association between the reported difficulties and the subsequent type of infant nutrition after hospital discharge. It is reasonable to consider that the relationship between the duration of hospital stay and the difficulties in breastfeeding after discharge is linked to the lack of skin-to-skin contact due to the hospital policy to prevent mother-child transmission of SARS-CoV-2 infection.

The impact of COVID-19 on public health has been dramatic, and thousands of research articles were published focusing solely on the impact of COVID-19 on maternal health, child health and nutrition. In the first months of the pandemic, vertical transmission of SARS-CoV-2 had been postulated [14]. This hypothesis raised concerns for pregnant women around the world. However, subsequent studies demonstrated that the probability of placental transmission is usually less than 5% [15] and that the virus cannot enter the mammary gland [16]; moreover, no live SARS-CoV-2 had been isolated in breastmilk, making it unlikely that breastmilk could be a vector for SARS-CoV-2 transmission [17,18]. Furthermore, other evidence showed the presence of SARS-CoV-2 IgA and IgG antibodies in the breastmilk of COVID-19 recovered women, raising the possibility that these antibodies could provide specific immunologic benefits to breastfeeding infants and provide protection against virus transmission [19,20].

Despite these reassuring data, guidelines and local practices that emerged during the pandemic were largely contrary to the promotion and protection of breastfeeding, including the restriction of parental presence at the bedside, complete separation of mothers who were either confirmed or suspected as COVID-19 positive from their infants, discouragement or abandonment of skin-to-skin contact and direct breastfeeding, early discharge following birth, and a lack of access to in-person pediatric follow-up and breastfeeding assistance [21]. Therefore, especially in the first phase of the pandemic, separation of the newborns from their mothers was applied in many settings, even though such a practice has been considered a violation of human rights [22].

In 2021, the RCOG, UNICEF, and WHO suggested that suspected or infected mothers should stay together with their infants after delivery, keep skin-to-skin contact and breastfeed directly with careful precautions if mothers feel well [23,24,25]. Precautions for direct breastfeeding put forward by the above organizations included washing hands before touching the infant, wearing a medical mask during any contact with the infant, and routinely cleaning and disinfecting surfaces that mothers had touched. Precautions for expressing breast milk included wearing a mask during expression, washing hands before touching any pumps/bottle parts and expressing breast milk, following recommendations for proper pump cleaning after each use, and feeding expressed milk to the infant by a healthy caregiver who was not at risk for COVID-19, if possible [26].

This is the first study to highlight the lack of healthcare information and support in breastfeeding women with SARS-CoV-2 infection, and our results are in line with previously published data on non-infected women during the pandemic. In particular, Costantini et al. demonstrated a significant reduction in healthcare support and advice for breastfeeding women during the pandemic [27], Ceulemans et al. showed that 50% of included pregnant women reported less counseling on breastfeeding compared to before the pandemic [28], Lambelet et al. showed that 47.5% of pregnant women and 36.2% of breastfeeding women reported the pandemic affected their interaction with healthcare services [29], while Barasel et al. demonstrated that a prenatal breastfeeding education program increased the breastfeeding motivation and decreased fears of breastfeeding in a population of non-infected women during the COVID-19 pandemic [30]. When considering the impact of SARS-CoV-2 infection on the breastfeeding practice, our results are in contrast with reported data from Rani et al., who evaluated the type of breastfeeding in a cohort of infected women. Included women were also separated from the baby at delivery according to the local practice, but all of them received specific support and information on hand and respiratory hygiene practices from the healthcare personnel [31]. According to these data, at the 8-week follow-up, 69% of women were practicing exclusive breastfeeding, while 23% of them used formula milk. Regarding our data, the difference in rates of exclusive and mixed breastfeeding may be due to the different reported rates of received information and support on how to reduce the risk of transmission of SARS-CoV-2 infection.

This is the first study to evaluate the psychological impact of SARS-CoV-2 infection on breastfeeding practice. Current data on non-infected breastfeeding women during the pandemic are controversial. Ceulemans et al. and Junker et al. did not observe a negative impact of the lockdown on self-reported breastfeeding practices [28,32]. On the other hand, Costantini et al. showed that 36.8% of pregnant women and 8.2% of breastfeeding women who answered their survey indicated that the pandemic had an impact on their current pregnancy experience or breastfeeding experience [27]. The difference between our data and previously published surveys on non-infected breastfeeding women may highlight the psychological impact of SARS-CoV-2 infection and the challenging condition of breastfeeding under the severe restrictions and precautions due to the infection.

This is also the first study to describe possible correlations between maternal factors, such as mode of delivery and duration of hospital stay, and the reported difficulties in breastfeeding among women with SARS-CoV-2 infection. Maternal COVID-19 has been reported to be associated with higher rates of cesarean section at the beginning of the pandemic, even if recent evidence has demonstrated that the standards of obstetrical assistance for pregnancies complicated by SARS-CoV-2 infection improved over time [33]. Furthermore, we also demonstrated that women experiencing difficulties in breastfeeding are more likely to feed their babies with infant formula after hospital discharge.

Limitations of our study include a small cohort of surveyed patients, the lack of pretesting the survey, the lack of information regarding the neonatal outcome, and the lack of standardized scales for the evaluation of the psychological impact of SARS-CoV-2 infection on breastfeeding, such as Perceived Stress Scale (PSS) or Edinburgh Postnatal Depression Scale (EPDS) to evaluate the role of breastfeeding in reduction of postpartum depression [34]. Furthermore, our cohort did not include women without SARS-CoV-2 infection, therefore we could not compare the perceived stress between infected and non-infected breastfeeding women during the pandemic.

## 5. Conclusions

Our survey highlights the importance of healthcare support and information on hygiene practices to avoid mother-to-child transmission of SARS-CoV-2 infection, in order to decrease the perceived stress related to breastfeeding for infected mothers under restrictions, especially in women undergoing cesarean section and with a long hospital stay. Improvement in maternal awareness may help to increase the rate of exclusive breastfeeding, enhancing the nutritional benefits and the long-term outcomes of newborns and women.

## Figures and Tables

**Figure 1 ijerph-20-00228-f001:**
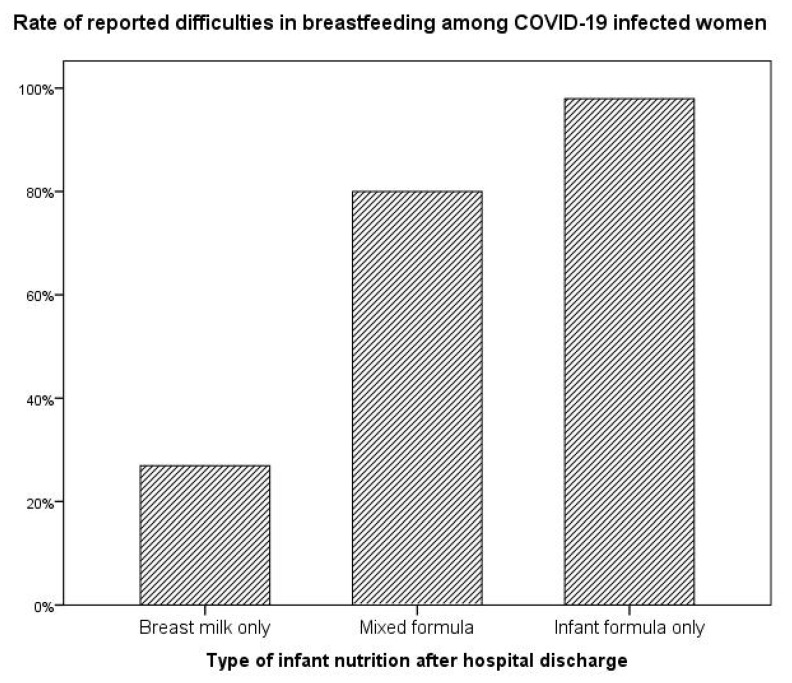
Infant nutrition after hospital discharge according to reported difficulties in breastfeeding.

**Table 1 ijerph-20-00228-t001:** Pregnancy characteristics of COVID-19 infected women.

Pregnancy Characteristics	*n* = 100
Maternal age (years)	30.1 ± 6.1
Marital status	
Unmarried	40 (40%)
Married	58 (58%)
Divorced/Separated	2 (2%)
Occupational status	
Student	3 (3%)
Housewife	50 (50%)
Employed	45 (45%)
Unemployed	2 (2%)
Parity	0.85 ± 1.1
Gestational age at delivery (weeks)	38.7 ± 1.9
Preterm birth	7 (7%)
Mode of delivery	
Vaginal delivery	35 (35%)
Cesarean section	65 (65%)
Hospital stay (days)	7.2 ± 3.5
Hospital stay > 5 days	57 (57%)
COVID-19 symptoms	27 (27%)

**Table 2 ijerph-20-00228-t002:** Breastfeeding characteristics of COVID-19 infected women.

Breastfeeding Characteristics	*n* = 100
Received information on hygiene practices	77 (77%)
Received advice on breast pumps	60 (60%)
Skin-to-skin contact at delivery	3 (3%)
Breastfeeding duration (months)	1.38 ± 2.12
Breastfeeding > 1 month	31 (31%)
Infant nutrition at hospital	
Exclusive breast milk	8 (8%)
Mixed formula	2 (2%)
Infant formula	90 (90%)
Infant nutrition at home	
Exclusive breast milk	26 (26%)
Mixed formula	25 (25%)
Infant formula	49 (49%)
Reported difficulties in breastfeeding	75 (75%)
Causes of difficulties in breastfeeding	
Reduced production of milk	18 (18%)
Fear of mother-to-child transmission	29 (29%)
Separation from the newborn	25 (25%)
Voluntary decision	3 (3%)
Reported impact of COVID-19 infection on breastfeeding	66 (66%)

**Table 3 ijerph-20-00228-t003:** Incidence of reported difficulties in breastfeeding and pregnancy characteristics in COVID-19 infected women.

Pregnancy Characteristics	Reported Difficulties in Breastfeeding		*p*	OR (95% CI)
	Yes (*n* = 75)	No (*n* = 25)		
COVID-19 symptoms	22 (29.3%)	5 (20%)	0.363	1.66 (0.55–4.98)
Nulliparous	38 (50.7%)	15 (60%)	0.418	1.46 (0.58–3.66)
Received information to reduce transmission	57 (76%)	20 (80%)	0.681	0.79 (0.26–2.41)
Received information on breast washing	9 (12%)	3 (12%)	1.000	1.00 (0.24–4.02)
Preterm birth	5 (6.7%)	2 (8%)	0.821	0.82 (0.15–4.52)
Cesarean section	56 (74.7%)	9 (36%)	<0.001	5.24 (1.99–13.79)
Days of hospital stay > 5	48 (67.6%)	9 (39.1%)	0.015	3.24 (1.22–8.59)

## Data Availability

The data that support the findings of this study are available from the corresponding author upon request.

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
