# Peer review of "Impact of COVID-19 on Breastfeeding among SARS-CoV-2 Infected Pregnant Women: A Single Centre Survey Study"

_ijerph, 2022, doi:10.3390/ijerph20010228_

Round 1
Author Response
Review report
A brief summary
A novel study that aimed to investigate the impact of COVID 19 on breastfeeding practices among women infected with SARS CoV-2.
Comments on key sections
Title: well written, self-explanatory
Introduction: Well written
Aim of the study, well stated (Ln66-67)
Materials and methods
Well written methodology, states study population, selection criteria, exclusion/inclusion criteria, anonymity.
We thank the reviewer for the considerations on our work
Minor suggestion: consider including:
1) How participants consents were obtained
2) pretesting of the questionnaire
1) Women were enrolled at admission to the Maternal Unit for the delivery and consent was asked and obtained before participating to the survey (Page 2 Lines 81-82)
2) we did not pretest the survey. We included the lack of pretesting in the Limitations (Page 8, Line 287)
Results: Commendable. Observation, table2 (Ln 167) and Table 1 (Ln 123) identical same except for different titles.
Please replace table 2(Ln 167) with the correct table content.
We thank the reviewer, we replaced the wrong Table 2.
Discussion: well written includes limitation of study
Conclusion: Suggest more emphasis on the findings on psychological impact of SARS-CoV-2 infection on
breastfeeding practice. Includes recommendation- good (Ln 284-284)
Minor change- consider including recommendation in the section title (Conclusions and Recommendation (Ln
278)
Reference: Ok
We thank the reviewer for the suggestion, we changed the Title of the last section
Reviewer 2 Report
General comment
Would Covid vaccination status be an additional useful variable?
Well written carefully crafted paper with a logical progression of information.
Clear concise and easy to understand.
Analyses simple but relevant.
Relation of results to the development of past research and to the development of new ideas in the dynamic and complex situation created by the COVID pandemic is illustrated well in the discussion.
How could the research findings feed into policy making?
Discussion
line 199: expand a bit and describe hospital stay length equated with lack of skin to skin contact
lines 255 - 264: perhaps include a comment on mothers that were multiparous and struggled with breastfeeding during COVID because they suffered from the negative impact of emotional stress /educational uncertainty from the medical profession during the pandemic. In other words in a proportion of the population, these negative impacts seemed to override previous pre-pandemic birth experience in individuals that previously had no problems with breast feeding.
line 274: A thought . There are computer based cognitive function tests that are fairly well established that have been used as a proxy for psychological stress. I think speed decision making is one of the first attributes to be affected. I am not a psychologist.
Khaligh-Razavi, S.M., Habibi, S., Sadeghi, M., Marefat, H., Khanbagi, M., Nabavi, S.M., Sadeghi, E. and Kalafatis, C., 2019. Integrated Cognitive assessment: speed and accuracy of visual processing as a reliable proxy to cognitive performance. Scientific Reports, 9(1), pp.1-11.
Author Response
General comment
Would Covid vaccination status be an additional useful variable?
We thank the reviewer for the interesting question. However, how cohort was enrolled from Nov 2020 to May 2021, when vaccines against SARS-CoV-2 were not available or were not recommended in pregnancy (Page 3 Lines 137-138)
Well written carefully crafted paper with a logical progression of information.
Clear concise and easy to understand.
Analyses simple but relevant.
Relation of results to the development of past research and to the development of new ideas in the dynamic and complex situation created by the COVID pandemic is illustrated well in the discussion.
We thank the reviewer for the kind considerations
How could the research findings feed into policy making?
In the Conclusion section (Page 8, Lines 296-301) we highlighted how improving healthcare support and information on hygiene practices to avoid mother-to-child transmission of SARS-CoV-2 infection could decrease the perceived stress related to breastfeeding for infected mothers under restrictions, especially in women undergoing cesarean section and with a long hospital stay.
Discussion
line 199: expand a bit and describe hospital stay length equated with lack of skin to skin contact
We thank the reviewer for the suggestion. At Page 7, lines 213-216 we described the link between the hospital stay lenght and the lack of skin-to-skin contact.
lines 255 - 264: perhaps include a comment on mothers that were multiparous and struggled with breastfeeding during COVID because they suffered from the negative impact of emotional stress /educational uncertainty from the medical profession during the pandemic. In other words in a proportion of the population, these negative impacts seemed to override previous pre-pandemic birth experience in individuals that previously had no problems with breast feeding.
We thank the reviwer for this consideration. At Page 5 Line 182 we added in the manuscript that, according to our data, there was no significant difference in perceived difficulties in breastfeeding according to the parity status. See also Table 3.
line 274: A thought . There are computer based cognitive function tests that are fairly well established that have been used as a proxy for psychological stress. I think speed decision making is one of the first attributes to be affected. I am not a psychologist.
Khaligh-Razavi, S.M., Habibi, S., Sadeghi, M., Marefat, H., Khanbagi, M., Nabavi, S.M., Sadeghi, E. and Kalafatis, C., 2019. Integrated Cognitive assessment: speed and accuracy of visual processing as a reliable proxy to cognitive performance. Scientific Reports, 9(1), pp.1-11.
We thank the reviewer for his suggestion. At present, speed decision making tests have not been evaluated yet in pregnant women to test for postpartum depression, and the suggested reference did not include pregnant or breastfeeding women, so we do not have any data to advice this test to evaluate pregnant or postpartum women in relation to our study.
Reviewer 3 Report
Review:
Dear authors, the idea of this manuscript is very good, but I have some comments to make:
1). The vulnerability of pregnant women to covid is more complex,see my manuscript where I had written what other authors described: at your paragraph 44 it would be better to add the anatomical and physiological changes during pregnancy…link Mitranovici, M.I.; Chiorean, D.M.; Oală, I.E.; Petre, I.; Cotoi, O.S. Evaluation of the Obstetric Patient: Pregnancy Outcomes during COVID-19 Pandemic—A Single-Center Retrospective Study in Romania. Reports 2022, 5, 27. https://doi.org/10.3390/reports5030027 in the intoduction.
Khan, D.S.A.; Hamid, L.R.; Ali, A.; Salam, R.A.; Zuberi, N.; Lassi, Z.S.; Das, J.K. Differences in pregnancy and perinatal outcomes among symptomatic versus asymptomatic COVID-19-infected pregnant women: A systematic review and meta-analysis. BMC Pregnancy Childbirth 2021, 21, 801.
2). Table 1 and 2 is the same....
3).Has any baby been infected with covid 19 at birth? Has any baby been infected through breastfeeding, or by handling the baby during breastfeeding? The question is not includedin the questionnaire...You talk here in paragraph 210-211 about the absence of the virus in breast milk but you do not talk about the respiratory transmission from mother to child.
4).The authors do not present a comparison between patients with covid infevtion and those without covid in the respective period of time, both subject to the stress of the pandemic.
Thank you.
Author Response
Dear authors, the idea of this manuscript is very good, but I have some comments to make:
1). The vulnerability of pregnant women to covid is more complex,see my manuscript where I had written what other authors described: at your paragraph 44 it would be better to add the anatomical and physiological changes during pregnancy…link Mitranovici, M.I.; Chiorean, D.M.; Oală, I.E.; Petre, I.; Cotoi, O.S. Evaluation of the Obstetric Patient: Pregnancy Outcomes during COVID-19 Pandemic—A Single-Center Retrospective Study in Romania. Reports 2022, 5, 27. https://doi.org/10.3390/reports5030027 in the intoduction.
Khan, D.S.A.; Hamid, L.R.; Ali, A.; Salam, R.A.; Zuberi, N.; Lassi, Z.S.; Das, J.K. Differences in pregnancy and perinatal outcomes among symptomatic versus asymptomatic COVID-19-infected pregnant women: A systematic review and meta-analysis. BMC Pregnancy Childbirth 2021, 21, 801.
We thank the reviewer for the suggestion. We included the suggested references at Page 2 Lines 44-49
2). Table 1 and 2 is the same....
We thank the reviewer, we replaced the wrong Table 2.
3).Has any baby been infected with covid 19 at birth? Has any baby been infected through breastfeeding, or by handling the baby during breastfeeding? The question is not includedin the questionnaire...You talk here in paragraph 210-211 about the absence of the virus in breast milk but you do not talk about the respiratory transmission from mother to child.
We thank the reviewer for the consideration. At Page 3 Lines 129-130 we stated that no neonatal infection was reported among all the babies born from the enrolled women.
4).The authors do not present a comparison between patients with covid infevtion and those without covid in the respective period of time, both subject to the stress of the pandemic.
Thank you.
We thank the reviewer for the consideration. Our survey was focused only on SARS-CoV-2 infected pregnant women, so we did not have a control group. We included the lack of a control group in the Limitations section (Page 8 Lines 292-294).
Round 2
Reviewer 3 Report
It looks much better !In my opinion it can be published.